# ICAM-reg: Interpretable Classification and Regression with Feature Attribution for Mapping Neurological Phenotypes in Individual Scans

**Cher Bass** [1]                                                CHER.BASS@KCL.AC.UK
[1] *King's College London*
**Mariana da Silva** [1]
**Carole Sudre** [1]
**Logan Z. J. Williams** [1]
**Petru-Daniel Tudosiu** [1]
**Fidel Alfaro-Almagro** [1]
**Sean P. Fitzgibbon** [2]
[2] *University of Oxford*
**Matthew F. Glasser** [3]
[3] *Washington University in St Louis*
**Stephen M. Smith** [2]
**Emma C. Robinson** [1]                                         EMMA.ROBINSON@KCL.AC.UK

**Editors:** Under Review for MIDL 2021

## Abstract

Feature attribution (FA), or the assignment of class-relevance to different locations in an image, is important for many classification and regression problems but is particularly crucial within the neuroscience domain, where accurate mechanistic models of behaviours, or disease, require knowledge of all features discriminative of a trait. At the same time, predicting class relevance from brain images is challenging as phenotypes are typically heterogeneous, and changes occur against a background of significant natural variation. Here, we present an extension of the ICAM framework for creating prediction specific FA maps through image-to-image translation.

**Keywords:** Interpretable, Classification, Regression, Deep Generative Networks.

## 1. Introduction

Brain images represent a significant resource in the development of mechanistic models of behaviour and neurological/psychiatric disease as, in principle, they capture measurable neuroanatomical traits that are heritable, present in unaffected siblings and detectable prior to disease onset. For many complex disorders, however, these features of disease are subtle, variable and obscured by a back-drop of significant natural variation in brain shape and appearance; this makes them extremely difficult to detect.

To detect features of disease in brain imaging, recent studies have started to apply deep learning methods that examine features or the weights of CNNs, called feature attribution (FA) methods. These methods include gradient based methods that analyse the gradients with respect to a given input image such as guided backpropagation (Springenberg et al.,

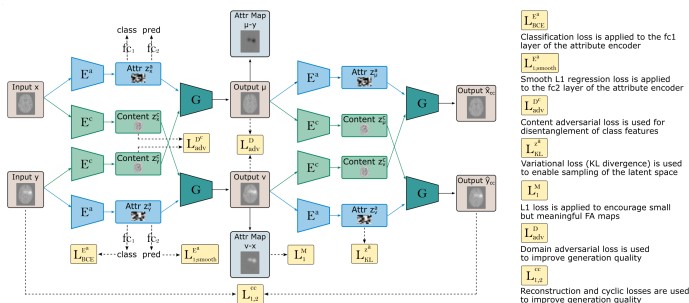

Figure 1: Overview of method. An example of how ICAM performs classification/ regression with attribute map generation for 2 given input images x and y.

2014), and perturbation methods such as occlusion. These methods however are low resolution, also do not detect heterogeneous structures (i.e. features that do not appear in all subjects), and often require averaging across multiple subjects to achieve good detection (Böhle et al., 2019). An alternative method called VA-GAN (Baumgartner et al., 2018) that uses a generative model, was still unable to detect all salient features.

To address these challenges, in Bass et al. (2020) we developed ICAM; which addressed this problem through disentangling class-relevant disease *attributes* (attr) from class-irrelevant shape *content*. In this way the method was able to generate much more accurate maps of cortical atrophy due to Alzheimer's. Here, we extend the approach with a regression module, to enable the network to do regression as well as classification.

## 2. Methods

The goal of the $ICAM_{reg}$ framework (Bass et al., 2020, 2021) is to perform classification (or regression) with simultaneous feature attribution, by training a VAE-GAN to swap the classes of input images: $x$, $y$; changing only the features of each image which are specific to the target phenotype. The design of the network is outlined in Fig. 1.

## 3. Results

Here, we show our results on the UK Biobank for age prediction (see Bass et al. (2021) for full experiments on 2 other datasets). In our regression experiments, we found that brain age prediction by $ICAM_{reg}$ ($2.20 \pm 1.86$ MAE) performs competitively relative to other deep learning methods trained on age prediction using the UK Biobank (reported test MAE scores of $2.14 \pm 0.05$ (Peng et al., 2021)). In addition, we give a highlight of our qualitative results with $ICAM_{reg}$ in Fig. 2, showing an example of outlier explanation and interpolation between 2 subjects. In Fig. 2 A), aged match subjects with one subject predicted as an outlier (subject 2, predicted=56, true=47 years), are used to demonstrate outlier detection. Evidence for the outlier prediction of subject 2 is presented through translating between the 2 subjects, indicating the presence of larger ventricles, hippocampal atrophy and cortical shrinking in subject 2. In Fig. 2 B), we demonstrate that the latent space is interpretable by linearly interpolation between the encoded attribute vectors (i.e. of 2 subjects with

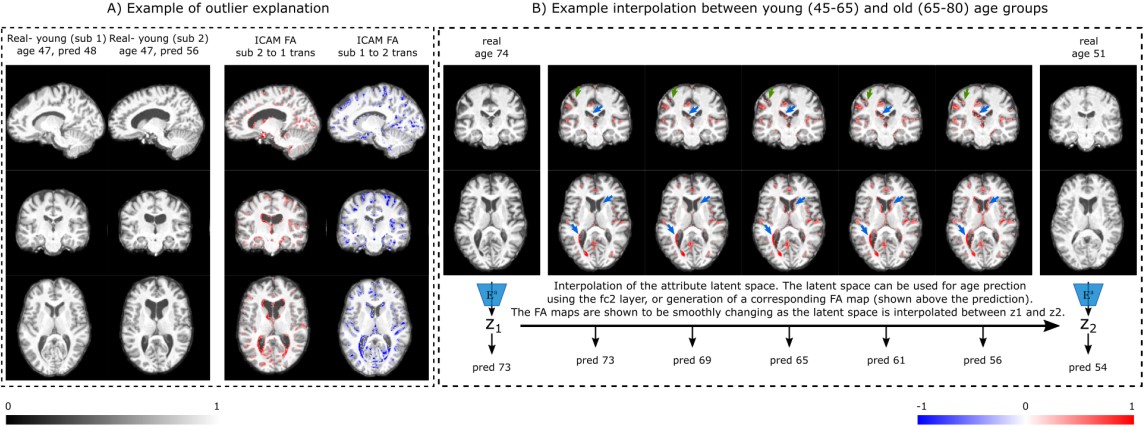

Figure 2: Highlights of UK Biobank results: outlier explanation, and interpolation between groups. Green arrows, cortex; blue arrows, ventricles.

different ages), and showing clear interpolation between them, where both the predicted ages and FA maps the are smoothly translated.

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
