# OpenReview forum: "ICAM-reg: Interpretable Classification and Regression with Feature Attribution for Mapping Neurological Phenotypes in Individual Scans"
_MIDL.io/2021/Conference/Short — MIDL 2021 Poster_

### Official Review · Reviewer_5KJp · 2021-04-25

**Confidence:** 3
**Final Rating:** 2

**Summary:**

The paper proposes an extension of the ICAM framework for creating Feature Attribution maps via an image-image translation task. Originally, the ICAM framework was designed to detect salient features by explicitly separating class level and class irrelevant content and tested on an Alziehmer's dataset to examine cortical atrophy (their 2020 paper) and was recently extended to include regression onto phenotypes(the 2021 paper). In this short paper extension, the authors re-examine age-prediction, which provides comparable performance with existing baselines. Their main contribution is to study their framework as applied to outlier explanation/prediction, and provide qualitative results for morphological changes observed by translating across different predicted ages.

**Strengths:**

1. The contribution and main problem statement of the paper, i.e. utilizing a FA classification/regression framework for outlier explanation, especially in the context of examining cortical atrophy in AD is interesting and possibly novel. The paper is also well written and concise, but doesn't cut corners.

2. The authors demonstrate that the ICAM provides similar performance on the age prediction task. Moreover, the qualitative results demonstrate that smooth interpolation of their FA maps could be a useful tool for examining the decisions made by the model, particularly in case of outliers. In my opinion, this is an interesting contribution towards developing explainable deep learning frameworks in general.

**Weaknesses:**

Some of the minor issues I had with the manuscript:

1. Given that the method itself combines several modules (Fig 1) and loss functions, I found the section describing the framework (architectural setup, training/testing setup) within methods to be inadequate in terms of explanation. I understand the space limitation, but it would really help make the manuscript a bit more self-explanatory.

2. Fig 2B. needs a bit more explanation. For example, it is unclear what the authors mean by 'groups' in this case. As per my understanding, the target variable (i.e. age) is continuous for the regression task. Do the authors group individuals of a certain age to perform the translation within the network? If so, how are these aggregated? What does 'smoothly translated' mean in this context?

**Deanonymize Review:**

no

**Detailed Comments:**

The main suggestions and points for clarification are:

1. The authors should consider providing a bit more explanation within methods for the setup. For example, the main loss terms/modules could be described within the white space surrounding Fig. 1, which is currently unused.

2. Could the authors please clarify the experimental setup of Fig. 2B? (Main points outlined within weaknesses)

**Justification Of The Rating:**

Although I did have a few points which I think need clarification, I did find the application interesting and novel. At the same time, there are some important aspects of the paper (mainly within the evaluation) that need addressing. Since the main contribution is more on the validation side than a methodological novelty, I would recommend a weak reject at this stage.

**Paper Type:**

both

**Special Issue:**

no

---

### Official Review · Reviewer_Xk6j · 2021-04-30

**Confidence:** 3
**Final Rating:** 3

**Summary:**

The article present an extension of ICAM an algorithm for feature attribution (FA), or the assignment of class-relevance to different locations in an image. The paper extends the usage of ICAM from classfication to regression tasks and  shows the application on the  UK  Biobank  for  age  prediction.

**Strengths:**

The application area of attributing feature relevance to different locations in an image is very important and the extension from the classification to the regression case also a very natural step forward.


**Weaknesses:**

The results are very unclearly described. Even if one is completely familiar with age regression from structural MRI, it's hard to follow what the actual results are showing. It's clear that ICAM performs  competitively  relative  to other  deep  learning  methods  trained  on  age  prediction  using  the  UK  Biobank, but the qualitative results are harder to understand. What is shown exactly int eh second panel of Figure 2 A)? And what is shown in Fig. 2 B)? The explanation y the single sentence "we demonstrate that the latent space is interpretable by showing clear interpolation between images of two different ages, where both the predictedages and FA maps the are smoothly translated." is very limited. Please consider reducing the space used for authors and adding some more context here.

**Deanonymize Review:**

no

**Justification Of The Rating:**

The algorithm itself is interesting and addresses an important problem, but the explanation both of the methods and results are lacking in detail, so the format of this short paper submission seems inappropriate for this.

**Paper Type:**

methodological development

**Special Issue:**

no

---

### Meta-Review · Program_Chairs · 2021-05-09

**Recommendation:** Accept (Poster)
**Confidence:** 4

**Metareview:**

Both reviewers highlight the importance of the topic and find the proposed approach interesting. Concerns are mainly on the lack of clarity about methods and results in the short paper. While I agree with that, it is somewhat less important for this short paper highlighting a recent submission (which is also available in full paper format), and I think this will make a nice contribution to the conference.
I encourage the authors to use the reviewer comments to improve clarity of the paper in the final version.

---

### Decision · Program_Chairs · 2021-05-11

Accept (Poster)